# Knowledge of Maternal Health Complications: A Critical Analysis Among Pregnant Women in Bangladesh

Rashida-E Ijdi[1,2]*, Chelsea M. Ducille[1], Kavita Singh[1,2]

**1** Department of Maternal and Child Health, Gillings School of Global Public Health, University of North Carolina at Chapel Hill, Chapel Hill, North Carolina, United States of America, **2** Carolina Population Center, University of North Carolina at Chapel Hill, Chapel Hill, North Carolina, United States of America

* ijdi@live.unc.edu

## Abstract

Despite notable reductions in Bangladesh's maternal mortality ratio (MMR), maternal deaths remain high, with many attributed to preventable obstetric complications. Timely recognition of maternal complications during pregnancy, delivery, or postpartum is crucial, yet maternal knowledge remains limited. This study explored pregnant women's knowledge of maternal complications in Bangladesh and predictors of knowledge acquisition. We analyzed data from 5,625 currently pregnant women using the nationally representative 2016 Bangladesh Maternal Mortality and Health Care Survey (BMMS). Knowledge of maternal complications was assessed based on the spontaneous mention of at least three danger signs across pregnancy, childbirth, or postpartum periods. Using a conceptual framework grounded in Andersen's behavioral model, multivariable Ordinary Least Squares (OLS) regressions were conducted to identify predictors of knowledge. Survey weights and clustering were accounted for in all analyses. Only 17.3% of pregnant women demonstrated sufficient knowledge of maternal complications. Knowledge varied significantly by age, education, and region. Women aged 30 and above, with secondary education or higher, and from urban areas were more likely to recognize key danger signs. Regional disparities were pronounced, with women in Chittagong and Sylhet divisions displaying significantly lower awareness compared to those in Dhaka. The study revealed low awareness of obstetric danger signs among pregnant women in Bangladesh, with substantial sociodemographic and regional disparities. Targeted interventions, particularly those focused on education, rural outreach, and antenatal care (ANC) counseling, are urgently needed to bridge the knowledge gap. Additionally, enhancing timely care-seeking may contribute to efforts to reduce maternal mortality in Bangladesh.

**Data availability statement:** This study analyzed publicly available, de-identified data from the 2016 Bangladesh Maternal Mortality and Health Care Survey (BMMS), conducted by the NIPORT, icddr,b, and MEASURE Evaluation. The dataset can be accessed through the UNC Dataverse at https://doi.org/10.15139/S3/X33NIZ for all qualified researchers. For long-term and stable data access, requests for additional details about the BMMS dataset or related metadata may be directed to the Data Access Committee, Carolina Population Center (CPC), University of North Carolina at Chapel Hill (email: dataverse@unc.edu; Phone: +1-919-962-3000).

**Funding:** This work was supported by the Carolina Population Center and its National Institutes of Health (NIH) Center for Scientific Review Grant number R24 HD050924, and by the United States Agency for International Development (USAID) through the Data for Impact (D4I) Associate Award number 7200AA18LA00008 (on which KS served as a co-investigator). The funders had no role in the study design, data collection and analysis, decision to publish, or preparation of the manuscript.

**Competing interests:** The authors have declared that no competing interests exist.

**Abbreviations:** ANC, antenatal care; BMMS, Bangladesh Maternal Mortality and Health Care Survey; CI, confidence intervals; MMR, maternal mortality ratio; VIF, variance inflation factors.

## Introduction

Worldwide, the maternal mortality ratio (MMR) has declined significantly, dropping 40% from 328 to 197 deaths per 100,000 live births between 2000 and 2023 [1]. Despite this progress, an estimated 260,000 women still die annually from pregnancy-related causes, which is equivalent to 712 deaths each day or one every two minutes [2]. The current global MMR is nearly three times higher than the target set by Sustainable Development Goal (SDG) 3.1, which aims to reduce the MMR to less than 70 per 100,000 live births by 2030 [2,3]. Maternal deaths are increasingly concentrated in the low- and lower-middle-income countries (LMICs), accounting for about 92% of all such deaths in 2023 [4]. This stark figure underscores how the vast majority of these preventable deaths disproportionately affect the world's poorest regions [4]. Sub-Saharan Africa and Southern Asia together accounted for approximately 87% of the estimated global maternal deaths, with Sub-Saharan Africa alone contributing about 70%, and Southern Asia around 17% - reflecting both their large population and persistent gaps in maternal healthcare [4]. Countries in Southern Asia have made significant progress, reducing their overall MMR from 405 in 2000–117 in 2023 - a 71% decline [2]. However, most remain far from achieving the SDG target [3]. As an LMIC, Bangladesh reduced its MMR from 400 per 100,000 live births in 2001–197 in 2016 [5]. While this improvement is commendable, the MMR in Bangladesh remains more than double the SDG goal, underscoring the need for continued efforts to accelerate maternal health gains.

Maternal mortality is influenced by a range of factors, including the availability, accessibility, and quality of maternal health services. Key interventions, such as skilled birth attendance, emergency obstetric care (EmOC), and antenatal care (ANC), have played a central role in reducing maternal deaths [2,5]. However, an often-overlooked determinant of maternal health outcomes is maternal knowledge and awareness of obstetric complications during pregnancy and childbirth. Knowledge of maternal complications is critical, as these are key indicators of potential risks during pregnancy, childbirth, and the postpartum period, necessitating urgent medical attention to protect the life of both mother and newborn [6,7]. It is estimated that around 15% of pregnant women experience serious obstetric complications, with key danger signs including heavy vaginal bleeding, swollen hands or face, and blurred vision during pregnancy; prolonged labor, seizures, excessive bleeding, and retained placenta during childbirth; and foul-smelling vaginal discharge and high fever during the postpartum period [2,6]. Many deaths from these maternal obstetric complications could be prevented if pregnant women and their families were able to recognize early warning signs and seek prompt medical care [7].

Knowledge of maternal complications directly impacts care-seeking behavior and timely use of maternal health services. Studies depict that timely recognition of danger signs and access to EmOC could prevent up to 74% of maternal deaths [8,9,10]. In resource-constrained settings, maternal mortality is often linked to the "three delays": delays in deciding to seek care, delays in reaching a healthcare facility, and delays in receiving appropriate treatment [11,12,13]. A key contributor to the

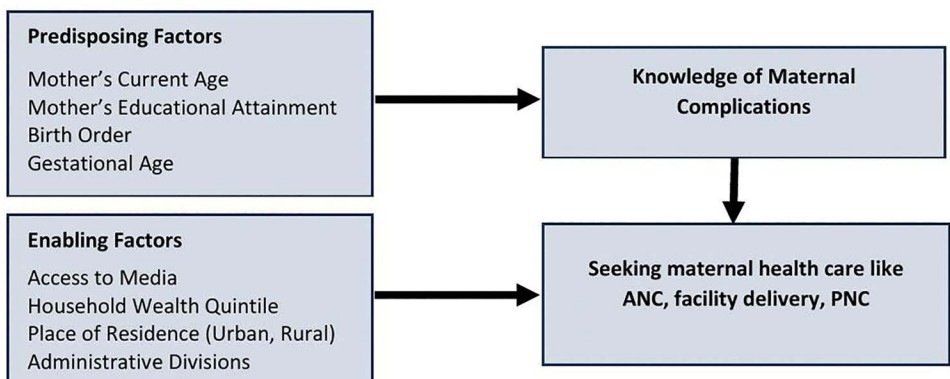

PLOS Global Public Health

first delay is a lack of knowledge of maternal complications, which directly impacts timely decision-making around seeking care [12]. Enhancing awareness of maternal complications, particularly for pregnant women, is critical during pregnancy, childbirth, and the postpartum period to ensure timely and life-saving interventions [12,13,14].

Knowledge of maternal danger signs remains alarmingly low in many LMICs, including Bangladesh. In rural Bangladesh, just one in four women could name three or more danger signs, highlighting a significant gap in knowledge on maternal complications [15]. The importance of maternal awareness is underscored by the fact that many leading causes of maternal mortality are preventable with timely recognition and appropriate action. Obstetric complications, such as postpartum hemorrhage and pre-eclampsia, can become fatal if not identified and managed urgently. Therefore, improving women's understanding of maternal health complications is a vital component of maternal mortality prevention and serves to complement the provision of quality clinical care. The World Health Organization (WHO) emphasizes that early identification of maternal complications, coupled with timely medical intervention, is a critical and most effective strategy for averting maternal deaths [16].

Given this context, enhancing women's knowledge of maternal complications is increasingly recognized as a priority for achieving maternal health targets. Bangladesh has an opportunity to leverage maternal education and awareness to sustain its progress in reducing MMR [5,15]. As the country strives to further lower its MMR and achieve the SDG target, understanding the level of knowledge regarding maternal complications and their determinants is vital. The objective of this study is to explore the persisting knowledge gaps in maternal health complications among pregnant women in Bangladesh. This study aims to inform policies and programs that enhance recognition of obstetric danger signs, encourage prompt care-seeking behavior, and support the reduction of maternal mortality in Bangladesh and similar settings by identifying existing information gaps and opportunities to strengthen maternal awareness as part of comprehensive strategies to end preventable maternal deaths.

## Methodology

### Conceptual framework

In this study, the conceptual framework for understanding knowledge of pregnancy complications among pregnant women was adapted from the framework proposed by Anderson et al. (1995) [17,18] to investigate the determinants influencing the utilization of obstetric services (Fig 1). By applying this conceptual framework, we assessed the awareness of obstetric danger signs among pregnant women regarding the necessity of obstetric services.

**Fig 1. Conceptual framework of factors associated with knowledge of maternal complications and health care-seeking behavior among pregnant women in Bangladesh.**

The modified conceptual framework groups the predisposing factors, including mother's current age, educational attainment, birth order, and gestational age, and enabling factors, such as household wealth quintile, access to media, place of residence (urban/rural), and administrative division. Predisposing factors represent individual-level characteristics that may contribute to a pregnant woman's knowledge of maternal complications and related care-seeking behaviors, while enabling factors reflect household- and community-level characteristics that may facilitate or constrain healthcare access of a pregnant woman [19]. Although the study cannot assess the impact off pregnant women's knowledge on maternal complications on seeking their maternal health care (such as antenatal care, facility delivery, postnatal care) because of the restrictions of the questionnaire.

## Data source

The study utilized data from the 2016 Bangladesh Maternal Mortality and Health Care Survey (BMMS), a nationally representative cross-sectional survey conducted between August 22, 2016, and February 10, 2017 [5]. The BMMS employed a multistage cluster sampling design to select households across all districts and administrative divisions of Bangladesh, targeting ever-married women aged 15–49 years and collecting data on maternal mortality with a three-year recall period (2014–2016) [5].

All data were fully de-identified before release, and the authors did not have access to any personally identifiable information at any stage of analysis. The original survey obtained ethical approval from relevant institutional review boards, and informed consent was collected from all participants by trained enumerators during data collection [5]. Further methodological details, including sample design, survey instruments, and data processing protocols, can be found in the 2016 BMMS report [5].

## Data analysis

The data inclusion criteria for this study limited the sample size to 5,625 for the currently pregnant women aged 15–49 years who answered knowledge questions on maternal complications during the survey.

**Outcome variable.** The outcome variable is 'knowledge of pregnancy complications' among currently pregnant women aged 15–49. They were asked the question, "Can you tell us what could be complications women may face during pregnancy, delivery, or postnatal period, for which she might even die?" The answer was categorized as severe headache, blurred vision, high blood pressure, edema/pre-Eclampsia, convulsion/eclampsia/unconsciousness, excess vaginal bleeding, foul-smelling discharge with severe fever, jaundice, tetanus, mal-presentation, prolonged labor, obstructed labor, delayed cord presentation, ruptured membrane, others, and don't know. Each category is coded as a binary variable, with '1' indicating awareness of the danger sign and '0' indicating lack of awareness. Considering prior studies in Bangladesh and other low and middle-income countries, in this study, women were classified as having 'sufficient knowledge' if they spontaneously mentioned ≥3 maternal complications across pregnancy, labor/childbirth, or postpartum [15,20,21,22]. Conversely, if she spontaneously mentioned <3 maternal complications (i.e., none, one or two), the respondent was coded as having 'low knowledge' [15,20,21,22]. This threshold reflects both established research practice and theoretical understanding that knowledge of multiple maternal complications is necessary for timely recognition of complications and appropriate care-seeking, as emphasized in WHO guidance [23].

**Covariates.** Based on the conceptual framework, covariates are included as pregnancy-related and sociodemographic factors. Pregnancy-related factors included are gestational age (categorized as 1st trimester, 2nd trimester, and 3rd trimester) and sociodemographic factors are birth order (0, 1, 2–3, 4 or more), mothers age (categorized as <20, 20–24, 25–29, 30 and more), mothers educational attainment (no education, primary incomplete, primary complete, secondary incomplete, and secondary complete or higher), access to any media (newspaper, radio and TV) at least once a week, wealth quintile (lowest, second, middle, fourth, and highest), place of residence (urban, rural), and administrative divisions (Barisal, Chittagong, Dhaka, Khulna, Mymensingh, Rajshahi, Rangpur, and Sylhet).

## Statistical analysis

In this study, descriptive analysis was performed to examine the distribution of currently pregnant women who participated in the study. Bivariate analyses were conducted to observe the distribution of each covariate by the outcome variable, knowledge of pregnancy complications. This was done for the binary variables "low knowledge" and "sufficient knowledge" and also for each complication. Among participants in the overall sample, the prevalence of knowledge was estimated using Rao-Scott χ2 tests. A model of the relationship between outcome variables and covariates was developed using multivariable Ordinary Least Squares (OLS) regressions employing a linear probability model for interpretability and comparability to calculate coefficients at 95% confidence intervals (CIs). Crude models were initially calculated in Model 1 to examine the bivariate relationship between knowledge and covariates. Adjusted coefficients were calculated in two final models: Model 2 included predisposing and enabling factors (excluding gestational age), while Model 3 included all covariates specified in the conceptual framework.

Given the large number of covariates, multicollinearity was assessed using Variance Inflation Factors (VIF) [24]. All covariates were acceptable with VIFs below 5, including our calculated VIF of 2.06. Given that there were no significant correlations between variables, all covariates were included in the final model.

All analyses were conducted using the statistical software package STATA version 19.0 [25]. Appropriate sampling weights for 2016 BMMS were applied using STATA's survey estimation procedure ("svy" command) to achieve nationally representative estimates of the population of Bangladesh after adjusting for sample strata and clusters [5,25].

**Ethics Statement.** This study is a secondary analysis of publicly available, de-identified data from the 2016 Bangladesh Maternal Mortality and Health Care Survey (BMMS), conducted by the National Institute of Population Research and Training (NIPORT), icddr'b, and MEASURE Evaluation. Ethical approval for the original survey was obtained from the relevant institutional review boards in Bangladesh, and all participants provided verbal informed consent prior to participation.

This study received ethical review and approval from the Institutional Review Board (IRB) at the University of North Carolina at Chapel Hill (UNC IRB Study No. 24–3136), dated December 17, 2024. A waiver of ethical approval was granted by the UNC Chapel Hill Office of Human Research Ethics, as the research involved secondary analysis of publicly available, fully anonymized data from the 2016 BMMS. The 2016 BMMS dataset, which contains no personally identifiable information, was accessed on September 19, 2023, through the UNC Dataverse upon request [26].

In accordance with UNC IRB policy, all data were securely stored and analyzed on password-protected devices with institutional endpoint protection. The dataset is publicly available, can be accessed through the UNC Dataverse upon request to all qualified researchers [26].

## Results

The 2016 BMMS surveyed 298,284 households, of which 321,214 women aged 15–49 were identified as eligible to participate. For this study, we focused on the 5,625 individuals who were pregnant at the time of the survey.

The descriptive statistics (Table 1) provide an overview of the 5,625 pregnant women included in the analysis. Only 17% of the respondents demonstrated sufficient knowledge of pregnancy complications. The distribution across trimesters showed a higher proportion in the second trimester (40%), followed by the first (27%), and the third trimester (33%). The majority of respondents resided in rural areas (74%), and most were from Dhaka (26%) and Chittagong (24%) divisions. Nearly 41% were experiencing their first pregnancy, with a large proportion of respondents being relatively young (34% aged 20–24, and 30% aged <20 years). Educational attainment varied widely; 23% of women had no formal education, while another 23% had completed secondary education or higher. The distribution across wealth quintiles was relatively even, ranging from 19% in the middle and highest quintiles to 22% in the fourth quintile, with the lowest and second quintiles each representing 20% of the respondents.

The distribution of knowledge of pregnancy complications is presented in Fig 2, highlighting that 36% of women recognized convulsion/eclampsia/unconsciousness as a significant obstetric risk. Awareness of other complications varies

**Table 1. Sociodemographic characteristics of currently pregnant women in Bangladesh, 2016 BMMS [N = 5,625].**

| Variables | Percentage (%) | Number (n) |
|---|---|---|
| **Knowledge of Pregnancy Complications** | | |
| **Sufficient Knowledge** | **17.3%** | **975** |
| **Low Knowledge** | **82.7%** | **4,650** |
| **Gestational Age** | | |
| 1st Trimester | 26.6% | 1,498 |
| 2nd Trimester | 40.1% | 2,254 |
| 3rd Trimester | 33.3% | 1,874 |
| **Birth Order** | | |
| 0 | 41.2% | 2,316 |
| 1 | 32.4% | 1,823 |
| 2 – 3 | 22.5% | 1,264 |
| 4 + | 3.9% | 222 |
| **Age (years)** | | |
| <20 | 29.5% | 1,662 |
| 20 – 24 | 34.1% | 1,921 |
| 25 – 29 | 23.2% | 1,304 |
| 30 + | 13.1% | 739 |
| **Educational Attainment** | | |
| No Education | 22.5% | 388 |
| Primary Incomplete | 6.9% | 826 |
| Primary Complete | 13.8% | 775 |
| Secondary Incomplete | 14.7% | 2,369 |
| Secondary completed or Higher | 22.6% | 1,268 |
| **Access to any media (newspaper, radio, TV)** | | |
| At least once a week | 58.4% | 3,283 |
| Less than once a week | 41.6% | 2,342 |
| **Wealth Quintile** | | |
| Lowest | 20.4% | 1,149 |
| Second | 20.0% | 1,128 |
| Middle | 18.9% | 1,061 |
| Fourth | 21.9% | 1,231 |
| Highest | 18.7% | 1,056 |
| **Place of Residence** | | |
| Urban | 26.3% | 1,477 |
| Rural | 73.7% | 4,148 |
| **Division** | | |
| Dhaka | 26.0% | 1,461 |
| Barisal | 5.3% | 300 |
| Chittagong | 24.0% | 1,351 |
| Khulna | 9.5% | 535 |
| Mymensingh | 8.2% | 460 |
| Rajshahi | 10.4% | 584 |
| Rangpur | 9.5% | 536 |
| Sylhet | 7.1% | 399 |

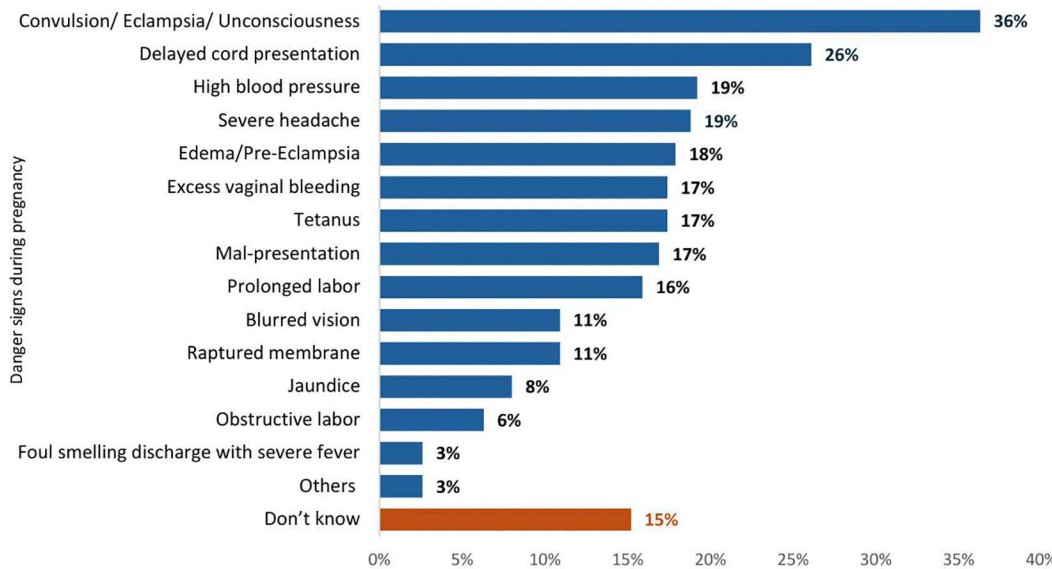

**Fig 2. Distribution of knowledge of maternal complications among currently pregnant women in Bangladesh, 2016 BMMS [N = 5,625].**

widely, with delayed cord presentation at 26%, high blood pressure, and severe headaches at 19% each. Results for knowledge of other complications include edema/pre-eclampsia (18%), excess vaginal bleeding, tetanus, and malpresentation (17.4%, 17.4%, and 16.9%, respectively), prolonged labor (16%), and blurred vision, ruptured membrane at 11% each. Less recognized issues include Jaundice (8%), obstructed labor (6%), and foul-smelling discharge with severe fever (3%). Notably, 15% of women are unaware of any pregnancy complications, indicating significant gaps in maternal health knowledge.

Table 2 presents the bivariate analysis of currently pregnant women in Bangladesh according to their level of knowledge on maternal complications, categorized as *low knowledge* (fewer than three complications identified spontaneously) and *sufficient knowledge* (three or more identified spontaneously). The results reveal notable sociodemographic disparities in maternal health complications knowledge by age, education, place of residence, and geographic location. Knowledge levels increased significantly with maternal age, 21% among women aged 30+, compared to 12% among those under 20 years, and with higher educational attainment (11% among women with no education compared to 27% among those with completed secondary or higher education. Women living in the urban areas (20%) and those with birth order 2–3 (20%) exhibit sufficient knowledge of maternal complications compared to their rural areas (16%) and first pregnancy (15%) counterparts. Economic status also had a significant association, with wealthier women showing higher awareness of maternal complications (25% in the highest wealth quintile, compared to 16% in the lowest quintile). Geographic disparities were evident, with lowest levels of knowledge observed in Chittagong (12%), and Sylhet (11%), compared to 22% in Dhaka division. A detailed analysis table (S1 Table) indicates that urban women were more likely to recognize maternal complications such as severe headaches (21% vs. 18%; $p = 0.022$) and high blood pressure (23% vs. 18%; $p < 0.001$), while rural women were more aware of tetanus (18% vs. 15%; $p = 0.037$). Women, aged 20 over, were more likely to identify high blood pressure and convulsions; whereas women aged 25 or older more often recognized excessive bleeding, prolonged labor, delayed cord presentation, and ruptured membranes compared to younger women under 20 years. Education played a significant role, with women having secondary education or higher demonstrated significantly greater awareness of severe

**Table 2. Bivariate analysis of the association between Knowledge of maternal complications and sociodemographic characteristics among currently pregnant women aged 15 – 49 years in Bangladesh, [N =5,625].**

| Variables | Knowledge of Maternal Health Complications | | p - value |
|---|---|---|---|
| | Low n (%) | Sufficient n (%) | |
| **Gestational Age** | | | |
| 1st Trimester | 1,251 (83.5%) | 250 (16.5%) | 0.174 |
| 2nd Trimester | 1,833 (81.3%) | 421 (18.7%) | |
| 3rd Trimester | 1,567 (83.6%) | 397 (16.4%) | |
| **Birth Order** | | | |
| 0 | 1,961 (84.7%) | 355 (15.3%) | 0.042 |
| 1 | 1,490 (81.7%) | 333 (18.3%) | |
| 2 – 3 | 1,018 (80.5%) | 246 (19.5%) | |
| 4 + | 181 (81.8%) | 40 (18.2%) | |
| **Age (years)** | | | |
| <20 | 1,461 (87.9%) | 201 (12.1%) | <0.001 |
| 20 – 24 | 1,580 (82.3%) | 341 (17.7%) | |
| 25 – 29 | 1,028 (78.8%) | 276 (21.2%) | |
| 30 + | 581 (78.7%) | 158 (21.3%) | |
| **Educational Attainment** | | | |
| No Education | 345 (89.0%) | 43 (11.0%) | <0.001 |
| Primary Incomplete | 713 (86.3%) | 113 (13.7%) | |
| Primary Complete | 668 (86.3%) | 106 (13.7%) | |
| Secondary Incomplete | 1,995 (84.2%) | 373 (15.8%) | |
| Secondary completed or Higher | 929 (73.3%) | 339 (26.7%) | |
| **Access to any media (newspaper, radio, TV)** | | | |
| At least once a week | 2,612 (79.5%) | 671 (20.5%) | <0.001 |
| Less than once a week | 2,039 (87.1%) | 303 (12.9%) | |
| **Wealth Quintile** | | | |
| Lowest | 970 (84.4%) | 180 (15.6%) | <0.001 |
| Second | 977 (86.7%) | 150 (13.3%) | |
| Middle | 883 (83.2%) | 178 (16.8%) | |
| Fourth | 1,028 (83.5%) | 203 (16.5%) | |
| Highest | 792 (75.0%) | 264 (25.0%) | |
| **Place of Residence** | | | |
| Urban | 1,185 (80.2%) | 293 (19.8%) | 0.013 |
| Rural | 3,466 (83.6%) | 682 (16.4%) | |
| **Division** | | | |
| Dhaka | 1,136 (77.8%) | 325 (22.2%) | <0.001 |
| Barisal | 240 (79.8%) | 61 (20.2%) | |
| Chittagong | 1,196 (88.5%) | 155 (11.5%) | |
| Khulna | 439 (82.0%) | 96 (18.0%) | |
| Mymensingh | 386 (84.0%) | 74 (16.0%) | |
| Rajshahi | 467 (79.9%) | 118 (20.1%) | |
| Rangpur | 433 (80.8%) | 103 (19.2%) | |
| Sylhet | 355 (88.9%) | 45 (11.1%) | |

headaches, high blood pressure, convulsions, and tetanus ($p<0.001$). Similarly, women from the highest quintile showed better understanding of maternal complications such as high blood pressure, edema, and convulsions.

Results of the multivariable regression analyses are presented in Table 3, illustrating the stepwise inclusion of covariates across the three models. **Model 1** presents crude (unadjusted) association, **Model 2** adjusts for socioeconomic and demographic factors, and **Model 3** adds pregnancy-related variable (e.g., gestational age). In **Model 1**, several factors were significantly associated with sufficient knowledge of maternal complications. Higher knowledge was observed among women with higher birth order (1st and 2–3), older maternal age (20–24, 25–29, and 30+years), secondary incomplete and secondary completed or higher education, and women belonging to the highest wealth quintile. Conversely, women residing in rural areas and those in the Chittagong, Mymensingh, and Sylhet divisions exhibited significantly lower knowledge compared to their urban and Dhaka counterparts.. After adjustment for predisposing and enabling factors in **Model 2** (birth order, maternal age, education, access to any media, wealth, residence, and division; excluding gestational age), maternal age and education remained robust predictors of knowledge. Older women continued to demonstrate greater awareness compared to those under 20 years, and the positive association with educational attainment persisted. Women with incomplete secondary education ($\beta=0.077$, $p=0.001$) and completed secondary or higher education ($\beta=0.181$, $p<0.001$) were significantly more knowledgeable than those with no education. Higher birth order also remained significant ($\beta=0.047$, $p=0.025$ for 2–3 births; $\beta=0.069$, $p=0.013$ for 4+births). The effect of wealth weakened after adjustment; only women in the second quintile had significantly lower knowledge compared to the poorest ($\beta=-0.038$, $p=0.020$). Regional disparities persisted, with women in Chittagong ($\beta=-0.105$, $p<0.001$), Mymensingh ($\beta=-0.047$, $p=0.044$), and Sylhet ($\beta=-0.099$, $p<0.001$) continuing to show significantly lower knowledge than those in Dhaka. **Model 3** included all covariates specified in the conceptual framework by adding gestational age, showed no substantial change in the results. Maternal age and education remained the strongest predictors of sufficient knowledge, exhibiting consistent positive gradients. Women aged 25–29 ($\beta=0.063$, $p=0.002$) and 30+years ($\beta=0.065$, $p=0.014$) were more likely to have sufficient knowledge than those under 20. Education maintained the largest effect size—women with completed secondary or higher education were approximately 18 percentage points more likely to demonstrate sufficient knowledge ($\beta=0.181$, $p<0.001$) compared to those with no education. Higher birth order (2–3) remained positively associated ($\beta=0.048$, $p=0.022$), while regional inequalities persisted, with Chittagong ($\beta=-0.105$, $p<0.001$), Sylhet ($\beta=-0.099$, $p<0.001$), and Mymensingh ($\beta=-0.048$, $p=0.040$) showing significantly lower knowledge relative to Dhaka. Gestational age itself was not significantly associated with knowledge of maternal complications.

The detailed multivariable regression analysis (S2 Table) identified significant predictors of knowledge of specific maternal complications. Older women demonstrated greater awareness than those under 20 years. Women aged 20–24 years were more likely to recognize severe headaches ($\beta=0.045$, $p=0.008$), high blood pressure ($\beta=0.062$, $p<0.001$), and convulsions ($\beta=0.059$, $p=0.005$), while those aged 25–29 years also showed higher awareness of excessive vaginal bleeding ($\beta=0.069$, $p=0.001$) and prolonged labor ($\beta=0.065$, $p=0.002$). Women aged 30 years and above had greater knowledge of high blood pressure ($\beta=0.078$, $p=0.002$) and convulsions ($\beta=0.093$, $p=0.007$). Education remained a strong and consistent predictor. Women with secondary incomplete education or higher were significantly more likely to recognize high blood pressure ($\beta=0.062$, $p=0.005$), convulsions ($\beta=0.125$, $p<0.001$), tetanus ($\beta=0.112$, $p<0.001$), and excessive bleeding ($\beta=0.073$, $p=0.002$). Those with secondary completed or higher education showed even stronger associations across several complications, including severe headache ($\beta=0.126$, $p<0.001$), convulsions ($\beta=0.209$, $p<0.001$), and tetanus ($\beta=0.165$, $p<0.001$). Access to media also played a significant role. Women exposed to television, radio, or newspapers were more likely to identify severe headaches ($\beta=0.052$, $p=0.015$), high blood pressure ($\beta=0.066$, $p<0.001$), and convulsions ($\beta=0.071$, $p=0.004$). The influence of wealth was weaker but positive, with only the highest quintile showing greater awareness of high blood pressure ($\beta=0.053$, $p=0.021$) and convulsions ($\beta=0.086$, $p=0.006$). Marked regional disparities persisted; women in Barisal were more knowledgeable, whereas those in Chittagong and Sylhet consistently reported lower awareness across complications. Mymensingh women showed mixed patterns, suggesting persistent geographic inequalities in maternal health knowledge across Bangladesh.

**Table 3. Multivariable regression analysis of factors associated with sufficient Knowledge of maternal complications among currently pregnant women aged 15 – 49 years in Bangladesh, [N = 5,625].**

| Covariates | Model 1 (Crude) | | Model 2 | | Model 3 | |
|---|---|---|---|---|---|---|
| | Coefficients | *p* - value | Coefficients | *p* – value | Coefficients | *p* - value |
| **Gestational Age** | | | | | | |
| 1st Trimester | 1.000 | – | – | – | 1.000 | – |
| 2nd Trimester | 0.022 | 0.148 | – | – | 0.019 | 0.177 |
| 3rd Trimester | - 0.001 | 0.931 | – | – | - 0.001 | 0.916 |
| **Birth Order** | | | | | | |
| 0 | 1.000 | – | 1.000 | – | 1.000 | – |
| 1 | 0.029 | 0.046 | 0.020 | 0.199 | 0.020 | 0.199 |
| 2 – 3 | 0.041 | 0.008 | 0.049 | 0.019 | 0.050 | 0.017 |
| 4 + | 0.028 | 0.372 | 0.075 | 0.052 | 0.074 | 0.018 |
| **Age (years)** | | | | | | |
| <20 | 1.000 | – | 1.000 | – | 1.000 | – |
| 20 – 24 | 0.056 | 0.000 | 0.038 | 0.012 | 0.038 | 0.011 |
| 25 – 29 | 0.090 | 0.000 | 0.060 | 0.003 | 0.060 | 0.004 |
| 30 + | 0.092 | 0.000 | 0.063 | 0.017 | 0.063 | 0.018 |
| **Educational Attainment** | | | | | | |
| No Education | 1.000 | – | 1.000 | – | 1.000 | – |
| Primary Incomplete | 0.026 | 0.254 | 0.041 | 0.080 | 0.041 | 0.084 |
| Primary Complete | 0.026 | 0.281 | 0.043 | 0.091 | 0.042 | 0.094 |
| Secondary Incomplete | 0.047 | 0.029 | 0.075 | 0.001 | 0.075 | 0.001 |
| Secondary completed or Higher | 0.157 | 0.000 | 0.1878 | 0.000 | 0.178 | 0.000 |
| **Access to Any Media (Newspaper, Radio, TV)** | | | | | | |
| At Least Once a Week | 1.00 | – | 1.00 | – | 1.00 | – |
| Less than Once a Week | -0.075 | 0.000 | -0.053 | 0.000 | -0.053 | 0.000 |
| **Wealth Quintile** | | | | | | |
| Lowest | 1.000 | – | 1.000 | – | 1.000 | – |
| Second | - 0.023 | 0.156 | - 0.051 | 0.002 | - 0.051 | 0.002 |
| Middle | 0.011 | 0.513 | - 0.045 | 0.022 | - 0.045 | 0.023 |
| Fourth | 0.008 | 0.642 | - 0.066 | 0.002 | - 0.065 | 0.002 |
| Highest | 0.093 | 0.000 | - 0.013 | 0.603 | 0.013 | 0.599 |
| **Place of Residence** | | | | | | |
| Urban | 1.000 | – | 1.000 | – | 1.000 | – |
| Rural | - 0.033 | 0.017 | - 0.008 | 0.584 | - 0.007 | 0.605 |
| **Division** | | | | | | |
| Dhaka | 1.000 | – | 1.000 | – | 1.000 | – |
| Barisal | - 0.020 | 0.490 | - 0.008 | 0.778 | - 0.009 | 0.743 |
| Chittagong | - 0.107 | 0.000 | - 0.099 | 0.000 | - 0.099 | 0.000 |
| Khulna | - 0.042 | 0.086 | - 0.037 | 0.118 | - 0.038 | 0.108 |
| Mymensingh | - 0.061 | 0.009 | - 0.044 | 0.057 | - 0.045 | 0.052 |
| Rajshahi | - 0.020 | 0.413 | - 0.005 | 0.830 | - 0.006 | 0.781 |
| Rangpur | - 0.030 | 0.205 | - 0.022 | 0.344 | - 0.023 | 0.317 |
| Sylhet | - 0.110 | 0.000 | - 0.091 | 0.000 | - 0.092 | 0.000 |

The multivariable analyses (Table 3 and S2 Table) highlight consistent patterns across both overall and complication-specific analyses. Maternal education and age emerged as the strongest and most consistent predictors of knowledge of maternal complications, with education exerting the largest independent effect even after adjusting for socio-economic and demographic factors. Wealth showed a weaker influence once education was considered, while regional disparities, particularly lower awareness in Chittagong and Sylhet divisions, remained substantial across models. These results highlight the critical role of education, age, and regional context in shaping women's understanding of maternal complications in Bangladesh.

## Discussion

The findings of this study reveal that knowledge of danger signs during pregnancy was low among the pregnant women in Bangladesh who participated in this nationally representative study. This underscores a crucial gap in the knowledge of maternal complications, as only 17.3% demonstrate sufficient knowledge, defined as the ability to spontaneously mention at least three danger signs during pregnancy, labor/delivery, or postpartum. This finding aligns with studies from other low- and middle-income countries, which similarly reported a lack of sufficient knowledge regarding pregnancy complications [21,27]. The observed deficiency of knowledge can be attributed to a range of sociodemographic factors, as indicated by our multivariable regression analysis. Key predictors of greater awareness encompass higher educational attainment, urban living conditions, and higher wealth quintiles. These results are consistent with prior studies emphasizing the importance of education and socioeconomic status as critical factors influencing health knowledge and behaviors [27]. These findings also support the conceptual framework adopted in this study [17].

Our analysis further revealed significant regional disparities in levels of knowledge of maternal complications. In particular, divisions like Chittagong and Sylhet showed significantly lower (approximately 10% lower) awareness compared to Dhaka. Sylhet and Chittagong have historically lagged behind Dhaka due to a combination of sociocultural norms (e.g., greater restrictions on women's mobility and decision-making), geographic and infrastructural challenges (hard-to-reach areas with limited transport), and weaker health system readiness and outreach compared to central divisions [5,28]. These findings suggest that regional health education programs and resources are unevenly distributed or less effective in certain areas. The Government of Bangladesh's ongoing sector-wide approach of the Health, Population, and Nutrition Sector Program (HPNSP) also highlights Chittagong and Sylhet divisions as hard-to-reach areas due to their unique health challenges stemming from geographical, cultural, and economic barriers [28]. The Government of Bangladesh is significantly prioritizing improving knowledge on maternal complications by strengthening the overall health system and the capacity of healthcare providers through birth preparedness and complication readiness, behavioral change communications, and counseling during ANC [28]. Despite these programs, the persistent and substantial knowledge gap regarding maternal complications among pregnant women indicates that the government initiatives are insufficient to adequately address this deficiency at the individual level for women of reproductive age in Bangladesh. While the HPNSP aims to close existing knowledge gaps and improve care-seeking practices to reduce maternal mortality [28], the observed disparity highlights the need for more effective or widespread implementation of its strategies. Tailored intervention efforts are required to enhance local health systems. Implementation of community-based health education is crucial for improving understanding and management of pregnancy-related health issues in these divisions and for aligning their healthcare outcomes with national standards.

This study underscores that knowledge on maternal complications in Bangladesh is significantly influenced by various socioeconomic factors. After adjusting for covariates, urban women, those with higher levels of education, and those from wealthier households were significantly more likely to identify key maternal complications such as convulsions, high blood pressure, and prolonged labor. These findings reinforce the critical role of ANC services as a platform for low-resource settings, especially in rural areas, and among women with lower education levels [5,15,29]. Ensuring ANC coverage and ensuring the inclusion of focused educational content on maternal complications could be instrumental in closing these knowledge gaps [15,29].

The persistently low knowledge on maternal complications calls for comprehensive and equitable health communication strategies. Interventions must prioritize rural populations, women with little or no education, and those from lower-income households. Tailored messaging through community health workers, improved counseling during ANC visits, and local media campaigns may help address these disparities [28]. Strengthening these efforts is essential to enhancing timely care-seeking and ultimately improving maternal health outcomes in Bangladesh.

## Strengths and limitations

This study has several notable strengths. First, the study utilized data from the 2016 BMMS, which employed rigorous sampling and data collection methods across all administrative divisions, allowing for high-quality, population-based estimates [5]. Despite being conducted several years ago, it remains the only large-scale survey in the country to systematically assess knowledge on maternal complications at the national level. This survey's comprehensive scope, methodological rigor, and enduring relevance make it a critical resource for identifying persistent knowledge gaps and guiding ongoing policy and programmatic efforts to improve maternal health outcomes [5]. Secondly, the large sample size of the survey enhances the statistical power and generalizability of the findings across diverse regions and population subgroups [29]. Thirdly, the study applies a robust conceptual framework that strengthens the interpretability of the results and their relevance to health behavior interventions. Fourthly, the analysis incorporates survey weights and clustering to account for the complex sampling design of the BMMS, ensuring nationally representative and unbiased estimates [5,29]. Finally, the findings of the study remain highly relevant for identifying priority areas for policy interventions and underscore the ongoing need to strengthen maternal health knowledge in Bangladesh.

This study also has some limitations. First, the data used in the analysis of this study are from the 2016 BMMS, which may not fully reflect the current state of maternal health knowledge among pregnant women in Bangladesh. Despite the age of the data, the 2016 BMMS remains the most recent nationally representative survey that comprehensively assessed maternal knowledge on maternal complications in Bangladesh. The large sample size and rigorous methodology of this survey offer valuable insights into the patterns and predictors of knowledge gaps that may persist or have evolved over time. Furthermore, we could not present data on antenatal care, which could be a strong predictor of knowledge on pregnancy complications [27], due to the absence of relevant information. Finally, the study's cross-sectional design limits the ability to draw causal inferences, and reliance on self-reported knowledge may be subject to recall or social desirability bias [27].

## Conclusion

This study showed a very low prevalence of knowledge on maternal complications among pregnant women in Bangladesh. Knowledge on maternal complications was influenced by several sociodemographic factors, with disparities observed across maternal age, education level, place of residence, and divisions. These findings collectively underscore the need for targeted interventions to bridge the knowledge gap that focuses on education and access to healthcare, particularly for younger, less educated, economically disadvantaged, and rural-women, to enhance maternal health outcomes across Bangladesh.

## Supporting information

**S1 Table. Bivariate analysis.**
(DOCX)

**S2 Table. Detailed multivariable regression analysis.**
(DOCX)

## Acknowledgments

The authors are grateful to the NIPORT, icddr,b, and the MEASURE Evaluation project for conducting the 2016 Bangladesh Maternal Mortality and Health Care Survey (BMMS) and for making the anonymized dataset publicly available. The authors also thank the United States Agency for International Development (USAID) for supporting the broader Data for Impact (D4I) initiative through which this research was conducted. The views expressed in this article are solely those of the authors and do not necessarily reflect the views of NIPORT, icddr,b, or the USAID. The authors appreciate the technical and administrative support provided by the Carolina Population Center at the University of North Carolina at Chapel Hill.

## Author contributions

**Conceptualization:** Rashida-E Ijdi, Kavita Singh.

**Data curation:** Rashida-E Ijdi.

**Formal analysis:** Rashida-E Ijdi, Chelsea M. Ducille.

**Funding acquisition:** Kavita Singh.

**Investigation:** Kavita Singh.

**Methodology:** Rashida-E Ijdi, Kavita Singh.

**Project administration:** Kavita Singh.

**Resources:** Rashida-E Ijdi.

**Software:** Rashida-E Ijdi.

**Supervision:** Rashida-E Ijdi, Kavita Singh.

**Validation:** Rashida-E Ijdi, Kavita Singh.

**Visualization:** Rashida-E Ijdi.

**Writing – original draft:** Rashida-E Ijdi.

**Writing – review & editing:** Rashida-E Ijdi, Chelsea M. Ducille, Kavita Singh.

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
