## [Decision Letter · Decision Letter 0]

8 Sep 2025

PGPH-D-25-01545

Knowledge of Maternal Health Complications: A Critical Analysis Among Pregnant Women in Bangladesh

Dear Dr. Rashida-E Ijdi,

Thank you for submitting your manuscript to PLOS Global Public Health. After careful consideration, we feel that it has merit but does not fully meet PLOS Global Public Health’s publication criteria as it currently stands. Therefore, we invite you to submit a revised version of the manuscript that addresses the points raised during the review process.

We look forward to receiving your revised manuscript.

Kind regards,

Ekta Saroha

Academic Editor

Journal Requirements:

Additional Editor Comments (if provided):

Reviewer #1:

Reviewer #2:

Reviewers' comments:

Reviewer's Responses to Questions

**Comments to the Author**

1. Does this manuscript meet PLOS Global Public Health’s publication criteria?

Reviewer #1: Yes

Reviewer #2: Yes

2. Has the statistical analysis been performed appropriately and rigorously?

Reviewer #1: Yes

Reviewer #2: Yes

3. Have the authors made all data underlying the findings in their manuscript fully available (please refer to the Data Availability Statement at the start of the manuscript PDF file)?

Reviewer #1: Yes

Reviewer #2: Yes

4. Is the manuscript presented in an intelligible fashion and written in standard English?

Reviewer #1: Yes

Reviewer #2: Yes

Reviewer #1: It is is nicely written paper which focuses on an important topic related to maternal health. The paper uses a huge nationally representative data set to examine the levels of knowledge on maternal complications and factors associated with the knowledge. I have some minor comments to address:

1. The title suggest it is a 'Critical analysis'; I do not see any justification why the authors think it is a 'critical' analysis.

2. The paper looked at pregnant women's knowledge about maternal complications, which is different from knowledge on obstetric danger signs (ODS). Knowledge of danger signs refers to a pregnant person's awareness of specific warning symptoms (like severe bleeding or reduced fetal movement) whereas knowledge of pregnancy complications is a broader understanding of the serious conditions that can arise during pregnancy, labor, or the postpartum period, such as preeclampsia or gestational diabetes. I suggest to use the term knowledge on maternal complications, not ODS.

3. Number of birth order (parity) is directly related to the age of the woman; putting both the variables in the model may give spurious results

4. I suggest to reduce the number of categories of education; no education and primary incomplete could be in one category, primary complete and secondary incomplete could be another category and secondary complete and higher could be another category. This will help to make concrete recommendation on the importance of education.

5. As the relationship of wealth quintiles varied between models, I suggest to group upper two quintiles and lower three quintiles together (40/60), to see whether there is any relationship with the wealth quintiles.

6. Do you have any other variables such as media exposure or exposure of any other source of information (like club membership), which can explain why some women have some understanding about maternal complications and some others do not?

Thank you.

Reviewer #2: Use of OLS Regression for Binary Outcome: The primary outcome (“sufficient knowledge” ≥3 ODS) is binary, yet OLS regression is used here. This requires justification as logistic regression would typically be preferred. If OLS is retained, you should explain why (e.g., interpretability, comparability) and discuss limitations.

Definition of Outcome Variable: The threshold of ≥3 signs to define “sufficient knowledge” appears subjective. Please provide citations or theoretical justification for this cut-off. Without justification, readers may question the validity of the outcome.

Results: For multivariable models, the manuscript currently lists β coefficients without interpretation. It would be good to translate some findings into percentage point differences or practical meaning such as “Women with higher education were xx percentage points more likely to recognize ≥3 signs compared to those with no education”.

Model Presentation: The description of Models 1 to 3 is not well-integrated into the results narrative. Currently it is unclear how the models build on each other. Kindly summarize briefly what each model adds and how key predictors change across models.

Regional Disparities: The results highlight regional differences (e.g., Sylhet vs. Dhaka) but the interpretation is limited. A stronger discussion of why these disparities exist (sociocultural, health system, service access) would strengthen the manuscript.

Terminology: Inconsistent use of terms (“pregnancy complications” vs. “obstetric danger signs”). Please choose one and use consistently throughout.

**Do you want your identity to be public for this peer review?** For information about this choice, including consent withdrawal, please see our Privacy Policy

Reviewer #1: No

Reviewer #2: **Yes: ** Mohammad Samim Soroush

---

## [Editor Report · Decision Letter 1]

14 Oct 2025

PGPH-D-25-01545R1

Knowledge of Maternal Health Complications: A Critical Analysis Among Pregnant Women in Bangladesh

Dear Dr. Rashida-E Ijdi,

Thank you for submitting your manuscript to PLOS Global Public Health. After careful consideration, we feel that it has merit but does not fully meet PLOS Global Public Health’s publication criteria as it currently stands. Therefore, we invite you to submit a revised version of the manuscript that addresses the points raised during the review process.

We look forward to receiving your revised manuscript.

Kind regards,

Ekta Saroha

Academic Editor
---

## [Editor Report · Decision Letter 2]

20 Oct 2025

PGPH-D-25-01545R2

Knowledge of Maternal Health Complications: A Critical Analysis Among Pregnant Women in Bangladesh

Dear Dr. Rashida-E Ijdi,

Thank you for submitting your manuscript to PLOS Global Public Health. After careful consideration, we feel that it has merit but does not fully meet PLOS Global Public Health’s publication criteria as it currently stands. Therefore, we invite you to submit a revised version of the manuscript that addresses the points raised during the review process.

We look forward to receiving your revised manuscript.

Kind regards,

Ekta Saroha

Academic Editor

Journal Requirements:

1. Please provide a detailed online Financial Disclosure statement. This is published with the article. It must therefore be completed in full sentences and contain the exact wording you wish to be published.

a) Please clarify all sources of financial support for your study. List the grants, grant numbers, and organizations that funded your study, including funding received from your institution. Please note that suppliers of material support, including research materials, should be recognized in the Acknowledgements section rather than in the Financial Disclosure. 

b) State the initials, alongside each funding source, of each author to receive each grant. For example: “This work was supported by the National Institutes of Health (####### to AM; ###### to CJ) and the National Science Foundation (###### to AM).”

c) State what role the funders took in the study. If the funders had no role in your study, please state: “The funders had no role in study design, data collection and analysis, decision to publish, or preparation of the manuscript.”

For more information, please go to our submission guidelines:

https://journals.plos.org/globalpublichealth/s/submission-guidelines#loc-financial-disclosure-statement

2. Please ensure that the funders and grant numbers match between the Financial Disclosure field and the Funding Information tab in your submission form. Note that the funders must be provided in the same order in both places as well.

3. Please update your online Competing Interests statement. If you have no competing interests to declare, please state: “The authors have declared that no competing interests exist.”

4. In this instance it seems there may be acceptable restrictions in place that prevent the public sharing of your minimal data. However, in line with our goal of ensuring long-term data availability to all interested researchers, PLOS’ Data Policy states that authors cannot be the sole named individuals responsible for ensuring data access (http://journals.plos.org/globalpublichealth/s/data-availability#loc-acceptable-data-sharing-methods).

5. We have noticed that you have uploaded Supporting Information files, but you have not included a list of legends. Please add a full list of legends for your Supporting Information files before or after the references list.
---

## [Editor Report · Decision Letter 3]

30 Oct 2025

Knowledge of Maternal Health Complications: A Critical Analysis Among Pregnant Women in Bangladesh

PGPH-D-25-01545R3

Dear Rashida-E Ijdi,

We are pleased to inform you that your manuscript 'Knowledge of Maternal Health Complications: A Critical Analysis Among Pregnant Women in Bangladesh' has been provisionally accepted for publication in PLOS Global Public Health.

Best regards,

Ekta Saroha

Academic Editor